# Human Whole Blood Interactions with Craniomaxillofacial Reconstruction Materials: Exploring In Vitro the Role of Blood Cascades and Leukocytes in Early Healing Events

**DOI:** 10.3390/jfb14070361

**Published:** 2023-07-11

**Authors:** Viviana R. Lopes, Ulrik Birgersson, Vivek Anand Manivel, Gry Hulsart-Billström, Sara Gallinetti, Conrado Aparicio, Jaan Hong

**Affiliations:** 1OssDsign AB, SE-754 50 Uppsala, Sweden; vl@ossdsign.com (V.R.L.); sg@ossdsign.com (S.G.); 2Department of Medicinal Chemistry, Translational Imaging, Uppsala University, SE-751 83 Uppsala, Sweden; gry.hulsart_billstrom@mcb.uu.se; 3Department of Clinical Science, Intervention and Technology, Division of Imaging and Technology, Karolinska Institute, SE-141 52 Huddinge, Sweden; ulrik.birgersson@ki.se; 4Department of Clinical Neuroscience, Neurosurgical Section, Karolinska University Hospital, SE-171 77 Stockholm, Sweden; 5Rudbeck Laboratory, Department of Immunology, Genetics and Pathology (IGP), Uppsala University, SE-751 85 Uppsala, Sweden; vivekanand.manivel@igp.uu.se; 6Department of Engineering Sciences, Applied Materials Science Section, Uppsala University, SE-751 03 Uppsala, Sweden; 7Faculty of Odontology, UIC Barcelona-International University of Catalonia, 08195 Barcelona, Spain; cjaparicio@uic.es; 8IBEC—Institute for Bioengineering of Catalonia, 08028 Barcelona, Spain

**Keywords:** biomaterials, human whole blood, coagulation, complement, calcium phosphate

## Abstract

The present study investigated early interactions between three alloplastic materials (calcium phosphate (CaP), titanium alloy (Ti), and polyetheretherketone (PEEK) with human whole blood using an established in vitro slide chamber model. After 60 min of contact with blood, coagulation (thrombin–antithrombin complexes, TAT) was initiated on all test materials (Ti > PEEK > CaP), with a significant increase only for Ti. All materials showed increased contact activation, with the KK–AT complex significantly increasing for CaP (*p* < 0.001), Ti (*p* < 0.01), and PEEK (*p* < 0.01) while only CaP demonstrated a notable rise in KK-C1INH production (*p* < 0.01). The complement system had significant activation across all materials, with CaP (*p* < 0.0001, *p* < 0.0001) generating the most pronounced levels of C3a and sC5b-9, followed by Ti (*p* < 0.001, *p* < 0.001) and lastly, PEEK (*p* < 0.001, *p* < 0.01). This activation correlated with leukocyte stimulation, particularly myeloperoxidase release. Consequently, the complement system may assume a more significant role in the early stages post implantation in response to CaP materials than previously recognized. Activation of the complement system and the inevitable activation of leukocytes might provide a more favorable environment for tissue remodeling and repair than has been traditionally acknowledged. While these findings are limited to the early blood response, complement and leukocyte activation suggest improved healing outcomes, which may impact long-term clinical outcomes.

## 1. Introduction

Today, the majority of reconstructive surgical procedures involve the use of various alloplastic materials, ranging from the rigid fixation of bone fractures using different titanium constructs to the filling of bony cavities with calcium-phosphate-based bone cement. The application of these diverse materials is dependent on their interfacial properties and subsequent interactions with cells and biological fluids, such as blood [1,2,3]. Post implantation, blood proteins and platelets immediately adhere to the material’s surface, consequently altering the blood clotting pathway. Coagulation occurs through surface-mediated reactions or tissue factor expression on cells [1,4,5]. Surface-mediated reactions (intrinsic pathway) occur via biomaterial surface interaction between coagulation factor XII (FXII) and prekallikrein, whereas tissue factor (extrinsic pathway) originates from proteins released from damaged tissue or expressed on activated immune (e.g., monocytes) and endothelial cells’ surfaces. The two pathways remain distinct until Factor X activation, which directs thrombin and fibrin production, culminating in clot formation [6,7]. Additionally, platelet activation is initiated by platelets adhering to the blood protein layer that develops on the biomaterial surface after implantation [1,6] Existing evidence suggests that specific coagulation factors and activated platelets are also involved in the activation of the complement system, [2,3,7,8], consequently creating a communication link between the coagulation and complement systems [4,5,9,10,11]. This crosstalk results in blood-mediated thromboinflammation, which has demonstrated beneficial effects on regeneration by promoting the homing of leukocytes near the biomaterial implant. This supports the immune cells, such as monocytes, in switching to pro-inflammatory (M1) or pro-regenerative (M2) phenotypes based on the cellular signals of the surrounding tissue under favorable conditions [12]. The nature and intensity of this acute response are crucial for promoting (or inhibiting) healing. Blood clotting and biomaterial interfacial properties significantly influence compatibility and tissue healing [13,14,15,16,17].

The pivotal function of blood in bone tissue regeneration is highlighted in a study by Thor [18], which demonstrates that blood clots on dental implant surfaces undergo conversion into bone. Nonetheless, the mechanism of action underlying the interaction between whole blood and biomaterials, particularly during the initial stages, has yet to be fully understood.

In this study, we focus on the initial stages of material–blood interaction by exposing three disc-shaped replicas of craniomaxillofacial implants constructed from polyether ether ketone (PEEK), titanium alloy (Ti), and triphasic calcium phosphate (CaP) to freshly collected human whole blood in an established in vitro slide model. The aim is to provide insights into how the early interaction between these materials and whole blood can contribute to their varying capabilities to promote in situ healing. 

## 2. Materials and Methods

### 2.1. Material Manufacturing

Medical-grade polyether ether ketone (PEEK) (Batch #6190394, ESSDE AB, Uppsala, Sweden), titanium (Ti) medical-grade 5, alloy Ti6Al4V (Lot # 01-958, Livallco stål AB, Stenkullen, Sweden), and triphasic calcium phosphate (CaP) (produced according to the process outlined by Gallinetti et al. [19]) were used. Each material was shaped into discs with a diameter of 16 mm and a height of 5 mm. The discs made from Ti and PEEK discs were cut from rods of the same diameter, polished with silicon carbide paper, and then subjected to an ultrasonic cleaning procedure to remove any residual particles. The discs underwent autoclaving for 20 min at a temperature of 121 °C (Autoclave Nüve OT 23B, Biotechlab, Sofia, Bulgaria). The CaP discs were prepared and supplied by OssDsign AB, Uppsala, Sweden. As experimental controls, polyvinylchloride (PVC) slides were used to ensure adequate heparinization and experimental reproducibility. The PVC slides were sterilized for 60 min at no less than 60 °C using 5% amoniumperoxidesulphate (APS).

### 2.2. Material Characterization

#### 2.2.1. Surface Morphology

The surface morphology of the materials was visualized using scanning electron microscopy (SEM, Zeiss Merlin, Oberkochen, Germany). To acquire high-quality SEM images, the following steps were performed. First, the samples underwent vacuum drying at 60 °C for 2 h prior to analysis to guarantee their complete dryness. Second, before observation, the samples were secured to the sample holder using carbon tape, silver tape was applied to the sides, and a Au-Pd coating was deposited on the surface to prevent charging during analysis. The sputter coating process utilized a Polaron SC7640 Sputter Coater (Thermo VG Scientific, Waltham, MA, USA) with a voltage of 2 kV and a current of 20 mA, operating for a duration of 60 s.

#### 2.2.2. Wettability

The materials’ hydrophilic or hydrophobic properties were assessed using water contact angle measurements, employing both sessile drop tests with ultrapure water and captive bubble methods. A macro contact angle meter (DM-CE1, Kyowa, Japan) with appropriate software (FAMAS, Kyowa, Japan) was used to perform the wettability tests and calculate the contact angles. The contact angle (θ) was defined as the angle between the solid phase and the liquid phase as described [20]. Prior to the contact angle experiments all materials were cleaned in water and dried in a desiccator. For the sessile method, a 2 µL drop of ultrapure water was dispensed on the surface of the tested sample. With the captive bubble method, the sample was immersed in water in a small container and a bubble of air was formed underneath the surface of the tested sample [21]. In both methods, the contact angles were calculated through the first 20 s of contact of the fluid with the tested surface to describe the dynamic wettability response. Three measurements were performed for each type of sample material.

### 2.3. Study Design

An in vitro slide chamber model developed by Hong et al. [22] was used to investigate the interaction between whole blood and the materials under investigation. In brief, we collected 1.3 mL of blood from seven healthy donors and transferred it to a two-well heparinized slide chamber with the same diameter as the prepared materials’ discs. Every material tested was subjected to the blood from each individual donor. The heparin-coating of the chamber surfaces allows blood contact to occur without artificial activation of the coagulation cascade. Subsequently, the materials were placed atop each well chamber pre-filled with human whole blood and secured with a clip. The closed chambers were then incubated for 60 min under rotation on a wheel at 20 rpm in an incubator at 37 °C.

At time point 0 min, 1 mL of fresh human blood was collected from each donor into ethylenediaminetetraacetic acid (EDTA) containing tubes at a final concentration of 4 mM, to serve as the baseline sample. Each experiment was carried out seven times using the two-well setup (i.e., fourteen surfaces tested per material) as depicted in Figure 1. None of the investigated materials underwent pre-treatment or received the addition of exogenous growth factors.

### 2.4. Heparinization and Blood Collection

The tubes and tips used in the blood experiments as well as the slide chamber were coated with the Corline heparin surface (Corline Systems AB, Uppsala, Sweden) according to the manufacturer’s recommendation. This resulted in a double-layered heparin coating, exhibiting a binding capacity of 12 pmol/ cm^2^ antithrombin, as previously described by Andersson et al. [23].

Human whole blood was collected from healthy adult donors who had abstained from taking any medication known to impact blood coagulation (e.g., ibuprofen and aspirin) for at least two weeks. During blood collection, the blood was partially heparinized through the use of 50 mL of Falcon tubes with 100 IU/mL heparin (LEO Pharma, Malmö, Sweden), resulting in a final concentration of 0.25 IU of heparin per ml of blood to partially inhibit blood coagulation. Blood was used within 20 min after collection at room temperature.

Informed consent was obtained from all blood donors prior to the experiment. Ethical approval was obtained from the regional ethics committee, with reference number 2008/264.

### 2.5. Processing of the Blood Samples

For each material, the fluid phase of the blood was collected into tubes containing ethylenediaminetetraacetic acid (EDTA) at a final concentration of 4 mM and subsequently processed for platelet count. The supernatants were analyzed with enzyme-linked immunosorbent assays (ELISA) for myeloperoxidase (MPO) and eosinophil peroxidase (EPX) release, coagulation, and complement and kallikrein–kinin markers.

Due to the porous nature of the CaP material, a reduction in the fluid phase was anticipated for the CaP material; therefore, all measured values were normalized based on the observed volume reduction. No pretreatment of the CaP material, such as saturation with saline solution or an equivalent, was performed prior to the study. This choice was made to mimic clinical use and prevent the introduction of potential confounding factors.

### 2.6. Analytical Procedures

#### 2.6.1. Macroscopic Visualization of Blood Interactions

Following 60-min incubation in the in vitro model, macroscopic evaluation of blood clotting was conducted for each material surface included in the study. The CaP discs were also sectioned using a scalpel to gross examine the internal structure. The results were captured in photographs.

#### 2.6.2. Platelet Count

Baseline samples were collected at 0 min and the residual whole blood was mixed with EDTA at a final concentration of 4 mM. Platelet count was determined with the use of a hematology analyzer (Sysmex XP-300^®^ Corporation, Kobe, Japan). Thereafter, the samples were centrifuged at 4500× *g* for 15 min at 4 °C to collect plasma for subsequent analysis. The plasma samples were stored at −70 °C until analysis and were measured in duplicate.

#### 2.6.3. Coagulation and Contact Markers

Thrombin–antithrombin (TAT) was analyzed quantitatively, as an indicator of coagulation activity. Anti-human thrombin pAb was used for capture and HPR-conjugated anti-human antithrombin pAb was used for detection (Enzyme Research Laboratories, South Bend, IN, USA). A standard pooled human serum diluted in working buffer served as a standard for TAT, in an in-house sandwich enzyme-linked immunosorbent assay (ELISA) for quantification as described earlier [24]. In summary, the capture antibodies were coated on Nunc Maxisorp ELISA plates (Thermo VG Scientific, Roskilde, Denmark) using phosphate buffer saline (PBS) and incubated overnight at 4 °C. Subsequently, the plates were blocked with 1% bovine serum albumin (BSA, Sigma Aldrich, Darmstadt, Germany) and incubated with samples for 60 min while shaking at room temperature. The plates were then washed with PBS-0.05% Tween (non-ionic surfactant), and biotinylated detection antibodies were added, followed by 60-min incubation. Subsequently, the plates were washed for 15 min with streptavidin–horseradish peroxidase (HRP). Tetramethylbenzidine (TMB, Surmodics, Eden Prairie, MN, USA) was then added until a bright signal was obtained, at which point the reaction was stopped by adding 1M H_2_SO_4_. Absorbance was measured via spectrometry at 450 nm.

#### 2.6.4. Myeloperoxidase (MPO) and EPX Release

The release of heme-containing enzymes, myeloperoxidase (MPO), and eosinophil peroxidase (EPX), was measured using MPO (Invitrogen, ThermoFisher Scientific, Vienna, Austria) and EPX (Mybioscource, San Diego, CA, USA) according to the manufacturer’s instructions.

#### 2.6.5. Complement Markers

The complement activation markers, the C3a fragment and soluble complexes C5b-9, were quantified using in-house sandwich ELISA as outlined in Section 2.6.3. Monoclonal antibody (mAb) 4SD17.3 and biotinylated rabbit polyclonal anti-C3a antibody (pAb) Rb-a-Hu were used for capture and detection, respectively. For the sC5b-9 ELISA, mAb anti-neoC9 (Diatec Monoclonals AS, Oslo, Norway) was used for capture, while anti-C5 pAb (Biosite BP373, Täby, Sweden) followed by SA-HRP (GE Healthcare, RPN1231V, Uppsala, Sweden) served as detection. Standards were prepared from Zymosan-activated serum calibrated against commercially available kits (MicroVue, Quidel Corp., Santa Clara, CA, USA).

#### 2.6.6. Kallikrein–Bradykinin (KK) Markers

Kallikrein in complex with the C1-inhibitor and anti-thrombin was quantified using an in-house sandwich ELISA. For the Kallikrein-C1 inhibitor (KK-C1Inh) and Kallikrein- anti thrombin (KK-AT) complex ELISA, sheep anti-human prekallikrein was used for capture and either biotinylated denatured anti-C1Inh antibody (alpha-antitrypsin purified) or HPR-conjugated anti-human antithrombin pAb was used for detection. Premade KK–C1inh complexes diluted in plasma were used as a standard for quantification.

### 2.7. Statistical Analysis

All statistical analyses were performed with Prism 9.4.1 (458) for Mac OS X, GraphPad Software Inc. (Boston, MA, USA). The results are expressed as mean + standard deviation (SD), unless stated otherwise. Outliers were identified using the nonlinear-regression-based method (ROUT) and removed. Subsequently, the statistical significance was calculated using a one-way analysis of variance (ANOVA), followed by Tukey’s post hoc test. The significance levels were set at *p* ≤ 0.05.

## 3. Results

### 3.1. Surface Topography Visualization

Scanning electron microscopy (SEM) unveiled clear differences in the surface topography among the studied materials (Figure 2A–C).

The CaP surface displayed a nanoporous architecture with crystal formations smaller than one micrometer (Figure 2A). In comparison, PEEK (Figure 2B) and Ti (Figure 2C) both demonstrated relatively consistent surfaces, similar to those found in commercial implants with only minor superficial grooves, attributable to the cutting and polishing processes.

### 3.2. Wettability

The wettability of the tested materials was measured using two different methods: sessile water drop and captive air bubble (Figure 3A,B).

CaP cement is a highly hydrophilic and porous material and thus, measurements of water contact angles of the cement with the sessile drop method were not possible, as anticipated. This is because the dispensed water drop was fully spread out on the CaP surface very quickly after contact (<1 s) and subsequently (in the next 1 s) absorbed in the bulk of the porous cement (Figure 3A).

Similarly, but unexpectedly, in this case, measurements of contact angle with the CaP cement using the captive air bubble method were not possible. This was because the delivered air bubble could not displace the water already in contact with the cement surface, which is strongly indicative of a highly hydrophilic surface (Figure 3A). The inability of obtaining quantitative values of water contact angles for the CaP material is noted as n.d. in the table shown in Figure 3B.

A surface is considered hydrophilic if the water contact angle is lower than 90° (sessile drop method). Under this definition, all tested materials are hydrophilic materials. CaP showed the highest hydrophilicity, as presented in the previous paragraph, followed by Ti and PEEK. The Ti surfaces were mildly hydrophilic (67.9 ± 1.7°) and the PEEK surfaces had water contact angles close to the hydrophilic limit (81.3 ± 1.4°). In the literature, the contact angle of the Ti alloy with different topographic finishes and untreated PEEK is between 30–70° and 70–90°, respectively [25,26].

### 3.3. Highest Coagulation to Ti

Similar to the conditions in an operating room where implant materials are exposed to the blood of a patient, this study used human whole blood. Following 60-min exposure to the surfaces at 37 °C in rotating chambers (as shown in Figure 1, in the Section 2), the discs of each material were inspected for coagulation reactions.

The macroscopic evaluation of the different material surfaces revealed strong activation and adherence of the blood cells and platelets. A dense blot clot was observed on the Ti surface, whereas PEEK and CaP displayed less dense blood clot formation (Figure 4A).

The residual platelet percentage indicates the level of platelet activation and adherence to the materials. Blood samples were obtained prior to exposure, and the initial number of platelets was determined. All materials demonstrated a significant platelet reduction compared to the initial quantity (Figure 4C). The CaP and Ti surfaces displayed comparable values which were significantly lower than PEEK. Blood clot formation for each material was confirmed by the generation of the thrombin–antithrombin (TAT) complex. All material surfaces initiated TAT, albeit at varying levels (Figure 4D). Ti caused significant production of TAT complexes (525 ug/mL ± 120) compared to the initial values or the other materials. CaP and PEEK showed low generation of TAT complexes but this was not significant compared to the initial amount (TAT levels of 77 ug/mL ± 10 for CaP and 192 ug/mL ± 54 for PEEK).

### 3.4. Highest Activation of Kallikrein–Bradykinin with CaP

The contact activation system is also linked to the kallikrein–bradykinin (KK) system. Here, the KK–AT complex was significantly induced by CaP (56 ± 9 nM), Ti (47 ± 8 nM), and PEEK (51 ± 10 nM) compared with the initial levels (Figure 5A). All materials exhibited an elevation in KK-C1INH production relative to the initial values; however, only CaP demonstrated a significant increase (144 ± 29 nM) (Figure 5B). Interestingly, the CaP surface showed the most pronounced increase in both complexes.

### 3.5. Highest Activation of Complement C3a and C5b-9 Complexes in CaP

The activation of the complement system was also significantly increased in the other materials, as shown by the values of C3a (497 ug/L ± 69) and sC5b-9 (304 ug/L ± 54) for Ti and C3a (461 ug/L ± 74) and sC5b-9 (237 ug/L ± 49) for PEEK, as shown in Figure 6A,B, respectively.

### 3.6. Highest Release of MPO with CaP

Leukocytes play an important role during blood coagulation and healing processes; hence, the activation of the most prevalent leukocytes and the release of their granule content were assessed. The focus was placed on two specific types of leukocytes: neutrophils and eosinophils, which are involved in the release of both human peroxidases, myeloperoxidase (MPO) and eosinophil peroxidase (EPX), respectively. Activation of both cell types and release of their granule content was confirmed by the activity of the human peroxidases MPO and EPX, as depicted in Figure 7.

CaP caused a significant release of MPO (34 ± 7 ng/mL), relative to the initial values, followed by a non-significant increase by Ti (18 ± 4 ng/mL) and PEEK (16 ± 3 ng/mL), as shown in Figure 7A.

All materials triggered the activation of eosinophils with a significant release of EPX compared to the initial values (Figure 7B); however, CaP and Ti exhibited comparable effects.

## 4. Discussion

In the present study, coagulation is most significantly activated by Ti and PEEK, whereas CaP appears to trigger blood cascades through complement activation. All materials activate leukocytes (EPX and MPO), with CaP inducing the highest neutrophil response (MPO). It has previously been assumed that (i) a strong trigger for coagulation and (ii) subsequent activation of platelets are essential to promote healing following the implantation of metals, alloys, and plastics [6]. These mechanisms were observed for Ti and PEEK; however, the blood response to the CaP material suggests that complement activation, rather than an initial strong coagulation and platelet activation, promotes tissue healing.

Ti exhibited the highest coagulation response of all materials studied (Ti > PEEK > CaP), as demonstrated by thrombin–antithrombin (TAT). CaP had notably low levels of the coagulation marker, TAT, despite an equal reduction in free circulating platelets in the blood as Ti at 60 min. Platelets were likely entrapped in the material’s porous structure, which is supported by the blood stain observed in the sectioned CaP material. This aspect is particularly important in the context of healing and repair processes, as CaP could function as a reservoir of growth factors, emphasizing the importance of the blood clot [13]. When confined within the pores, platelets could gradually release their content over time, which encompasses a variety of growth factors such as TGF−β, PDGF, VEGF, and IGF [27]. These growth factors have been shown to influence both angiogenesis and the differentiation of osteoprogenitor cells, promoting healing [28]. Upon activation, platelets have also been shown to regulate the production of monocyte cytokines that counteract inflammatory responses [29].

Beyond porosity, surface wettability influences platelet adherence and, consequently, blood clot formation [6,30,31]. Wettable surfaces (i.e., hydrophilic) are generally more favorable to interactions with blood [32]. This largely accounts for the increased clot formation on Ti and the smaller clot found on PEEK along with the higher and lower release of TAT, respectively, as anticipated given the surface wettability of the two materials. Prior findings by Hong et al. support our data demonstrating elevated levels of TAT associated with Ti surfaces [24,33]. Increasingly hydrophilic Ti surfaces can upregulate the deposition of the fibrin matrix, leading to more substantial blood clots [24]. In the context of bone healing, the hydrophilicity of Ti surfaces has been shown to promote proliferation and osteoblast precursor differentiation, as well as positively regulate angiogenesis, bone mineralization, and bone remodeling [34]. Interestingly, we found less clotting on the CaP material compared to Ti despite its high hydrophilicity. This outcome suggests that factors beyond wettability such as topography, charge, and intrinsic chemical composition influence blood coagulation, especially for CaP materials.

Another blood cascade tightly interlinked with the coagulation systems is the kallikrein–kinin system or the so-called contact activation system [35]. Kallikrein production is regulated by two complexes, kallikrein (KK)–AT and –C1INH, which are ubiquitously present in plasma. Kallikrein has a dual role: first, it interacts with coagulation elements that activate it, and second, it generates bradykinin in the kallikrein–kinin system. Bradykinin promotes tissue regeneration and stimulates the migration of various cell types, including neutrophils, fibroblasts, and endothelial cells among others [36,37,38,39]. In our study, CaP exhibited the highest production of both complexes, followed by PEEK and Ti with no significant upregulation observed for Ti or PEEK regarding KK–C1INH complexes.

The complement activation was markedly more pronounced with CaP, followed by Ti and PEEK as indicated by a significant increase in the production of both C3a and C5a (evidenced by sC5b-9) complement proteins. These findings align with those reported by Klein et al. [40], who demonstrated cleavage of C3 with calcium phosphate powders. Given the reciprocal relationship between contact and complement system activation as shown by Huang et al. [3], the significant complement system activation could explain the low values of the coagulation marker TAT seen for CaP, where fewer protein triggers occur in the clotting cascade. Furthermore, complement system activation has previously been linked with enhanced healing following trauma. The proteolytic cleavage of complement proteins C3 and C5 induces a diverse range of cellular responses, such as chemotaxis, cell activation, and cell adhesion. In preclinical models, C5aR activation by C5a induced strong chemotactic activity in osteoblasts [41]. Bone cells express complement components, including C3, C5, and related receptors, and are responsive to C3a and C5a. During bone healing, these molecules can promote osteoclast formation in a pro-inflammatory environment. Additionally, C5a promotes strong chemotactic activity in osteoblasts [2,42,43,44,45].

For instance, Ehrnthaller et al. [41] showed that mice deficient in C3 and C5 exhibited significantly reduced bone formation during the early healing phases, highlighting the complement systems’ importance for successful healing. The calcium phosphate (CaP) used in this investigation has demonstrated the ability to undergo resorption and replacement by bone during the healing process in both preclinical and clinical settings. A considerable presence of osteoblasts has been observed at early and late stages in preclinical models. Interestingly, minimal osteoclast activity was observed [19,20,46,47,48]; instead, a moderate to substantial presence of material-filled macrophages was observed, which are also responsive to the complement proteins C3a, C5a, and bradykinin [49]. Contrarily, titanium meshes [19], solid titanium implants, and PEEK have been found to trigger a foreign body reaction, leading to chronic inflammation that promotes fibrous encapsulation rather than bone formation [19,25,50].

Furthermore, the complement system is known for its ability to induce immune cell activation. Leukocytes, such as neutrophils and eosinophils, express receptors for complements of C3a, C5a, and other complement molecules [49]. Notably, neutrophils have more C5a receptors, while eosinophils possess a higher density of C3a receptors [51].

In response to various stimuli, leukocytes can release a variety of granule proteins, which includes hemeprotein myeloperoxidase (MPO) from neutrophils and human eosinophile peroxidase (EPX) from eosinophils [52,53]. In this study, the CaP induced a significant release of both MPO and EPX relative to the baseline values. Concurrently, both Ti and PEEK induced an increase in MPO and EPX levels, although only the elevation of EPX was statistically significant. In a study by Burkhardt et al. [17] neutrophils by releasing MPO and reactive oxygen species (ROS) were found to facilitate the initial provisional fibrin matrix deposition, as well as the secretion of growth factors and cytokines by the first wave of invading cells, such as blood cells.

Both MPO and EPX peroxidases can stimulate fibroblasts and osteoblasts to migrate, secrete collagen for normal tissue repair, and exhibit angiogenetic properties such as the VEGF factor [54,55,56]. Additionally, MPO and EPX inhibit osteoclast differentiation and consequently bone resorption [57,58].

Furthermore, MPO and EPX exhibit notable antibacterial properties. Myeloperoxidase (MPO), the primary component of neutrophil granules, plays a significant role in creating and maintaining an alkaline environment. This environment is essential for restricting bacterial proliferation and effectively fighting infections. Neutrophils account for 60 to 70% of immune cells in the blood, emphasizing their importance in this context [57,59].

Both neutrophils and eosinophils possess bactericidal activity through the production of reactive oxygen species (ROS). The ROS generated by EPX can attack and damage various bacterial components, including proteins, lipids, and nucleic acids, ultimately leading to bacterial cell death. By damaging and killing bacteria, EPX plays a vital role in controlling and preventing infections, supporting the immune system’s ability to clear invading pathogens [60].

The results of our investigation, emphasizing the antimicrobial characteristics of these molecules, may bear clinical relevance. These findings could provide insights regarding the relatively lower infection rates associated with the use of CaP implants in craniomaxillofacial reconstructive procedures [61,62].

Macrophages play a crucial role in the immune system, tissue repair, and regeneration. Originating from circulating monocytes, macrophages participate in multiple stages of the tissue healing process, ranging from the initial inflammatory response to tissue remodeling. It is essential, in the context of tissue regeneration, for macrophages to transition from the pro-inflammatory M1 phenotype to the anti-inflammatory M2 phenotype. The M1 macrophages are instrumental in the initial inflammatory response to injury or infection, assisting in eliminating pathogens and damaged tissue. However, a prolonged M1 response may result in chronic inflammation and subsequent tissue damage. The polarization of macrophages into an anti-inflammatory M2 phenotype helps ensure that the initial inflammatory response is followed by a resolution of inflammation and the facilitation of tissue repair and regeneration. Dysregulation of macrophage polarization can lead to impaired healing, chronic inflammation, or excessive scarring [63,64,65].

In addition, monocytes and macrophages, which are essential during inflammation, also respond to the complement proteins C3a, C5a, and bradykinin [51,66]. Considering the significant upregulation of C3a especially, as well as C5a in particular, eosinophils are likely to accumulate at the material surfaces, especially for CaP, which showed the most significant activation [66]. Eosinophils have been found to induce macrophage polarization from M1 into M2 [57,59], thereby playing a vital role in modulating the pro-inflammatory milieu into a regenerative environment. PEEK implants generally elicit chronic inflammation (M1) that causes fibrous encapsulation rather than bone formation and integration [63,64]. Conversely, Ti seems to be able to stimulate the body’s response in various ways depending on the anatomical location and distance to native bone. In a non-osseous environment or far from the bone defect (e.g., critical size defects), it generally causes fibrous encapsulation with no or minimal bone formation, whereas places in a bony cavity, such as in dental applications, are fully osteointegrated over time [64]. The CaP composition studied here has been shown to have a higher healing capacity in craniomaxillofacial reconstruction where it has been shown to be resorbed and replaced by bone during the healing process both in preclinical and clinical settings without signs of chronic inflammation [19,20,46,48,67,68,69,70]. This response seems to be clearly coupled with the material–macrophage interaction over time [19,20,46,47]. Taken together we, therefore, propose that both the complement and contact system activating properties of this CaP with blood at implantation initiate both the activation and recruitment of leukocytes and osteoblasts and inhibit osteoclasts at the site of implantation.

Considering that implantable materials are temporarily exposed to blood, it is highly probable that the complement activation is transient and subsides during the first days of implantation, potentially resulting in reduced neutrophil recruitment. Schmidt-Bleek et al. [71], who studied immune cell subpopulations of a bone hematoma, found that the composition of neutrophils exceeded that of peripheral blood 60 min post-surgery. Nonetheless, the neutrophil numbers returned to levels comparable to those in the circulating blood after 4 h. While neutrophils play an important role in the initial response to implantable materials, it is equally important to regulate this response over time as prolonged recruitment of active neutrophils may extend the initial beneficial inflammatory response and ultimately impair tissue regeneration [58,72].

In summary (Figure 8), this study revealed that upon short contact with circulating human whole blood (i) CaP primarily activated the complement system and leukocytes instead of coagulation cascades, (ii) Ti induced coagulation and fewer other blood cascades and leukocytes, and (iii) PEEK exerted an intermediate influence on all studied systems. These findings suggest an inverse activation relationship between the coagulation system and the complement system during the early stages with CaP and Ti materials.

The limitations of the study are the lack of inclusion of the tissue factor (extrinsic pathway) and that only one early time point was studied. Furthermore, this study only includes two types of white cells, neutrophils and eosinophils, as a result of the complement activation outcomes. Nevertheless, the primary objective of the study is to investigate the early interactions that arise from the contact between biomaterials and whole blood. Future studies should aim to explore how the inflammatory response to implantable materials is modulated over time, concentrating on the recruitment of various specific cells and their differentiation at various time points.

## 5. Conclusions

In the current study, coagulation was notably activated by Ti and PEEK, while with CaP, the complement is the main blood cascade triggered. Contrary to previous concepts emphasizing strong coagulation and platelet activation for healing following the implantation of metals, alloys, and plastics, the blood response of CaP material suggests that complement activation may be crucial for tissue healing as an initial event. These findings offer insights into blood–material interactions and could guide the further development and optimization of existing implantable materials used in today’s clinical practice. By modulating the initial immune response, the healing process could be optimized, possibly resulting in fewer complications and better osteointegration of implant materials. 

## Figures and Tables

**Figure 1 jfb-14-00361-f001:**
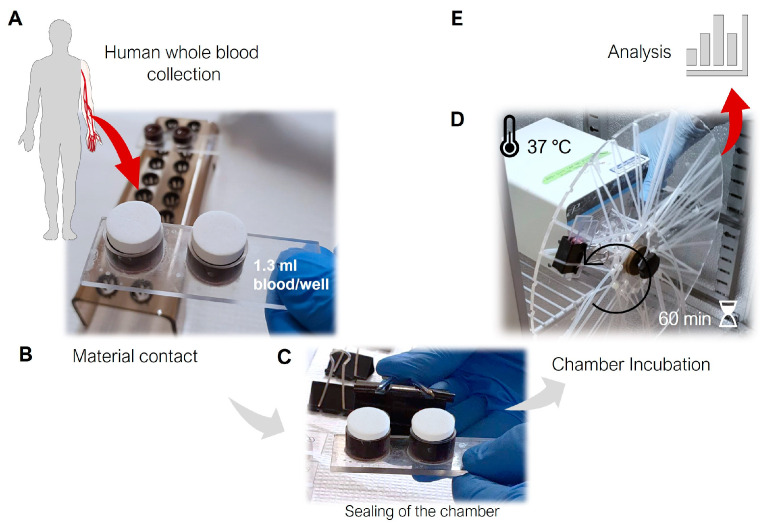
Schematic design setup of the whole blood chamber model. (**A**) Blood was drawn from healthy human donors, followed by the addition of 1.3 mL of fresh blood to each well of 2-well blood chambers. (**B**) Two samples of each material were introduced to the 2-well chambers just prior to (**C**) securely sealing the chambers using heparinized ethylene propylene o-rings and a clip. (**D**) Thereafter, the chamber was positioned on a rotating wheel at 37 °C for 60 min. (**E**) One mL of blood from each well was immediately collected and analyzed both before and after incubation, while the remaining blood was centrifuged, rapidly frozen, and stored at −80 °C for subsequent analysis.

**Figure 2 jfb-14-00361-f002:**
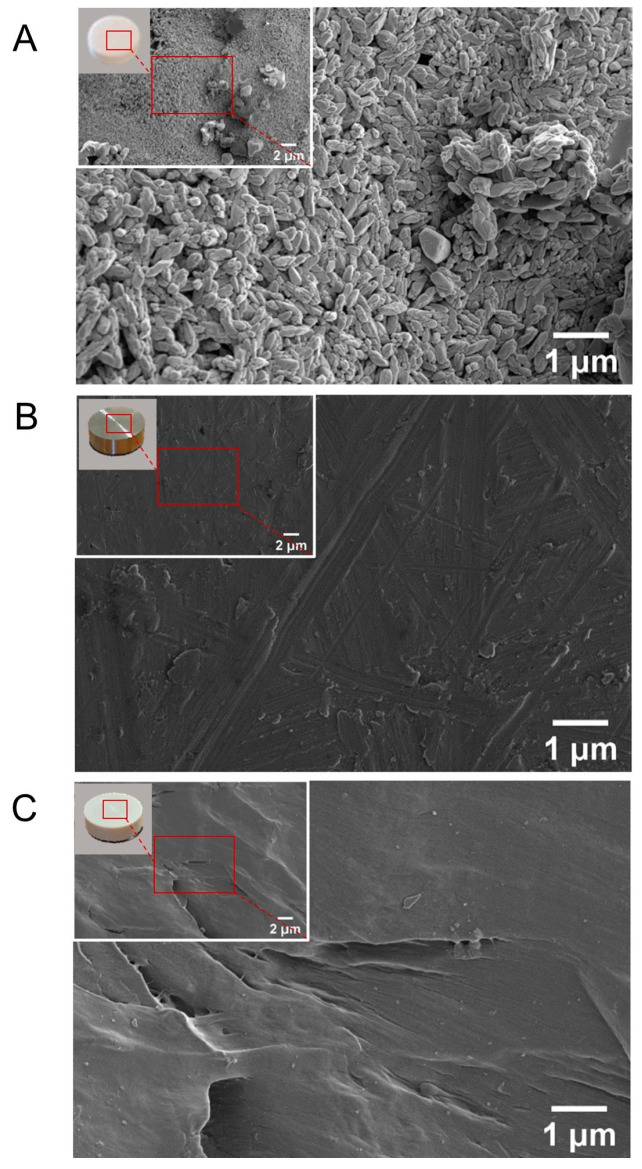
Scanning electron microscopy (SEM) images of the (**A**) calcium phosphate (CaP), (**B**) titanium alloy (Ti), and (**C**) polyetheretherketone (PEEK) disc surfaces examined in this study. Magnification of the top-left SEM images—10.00 kx for (**A**,**B**), and 5.00 kx for (**C**). Magnification of the central images—30.00 kx for (**A**–**C**).

**Figure 3 jfb-14-00361-f003:**
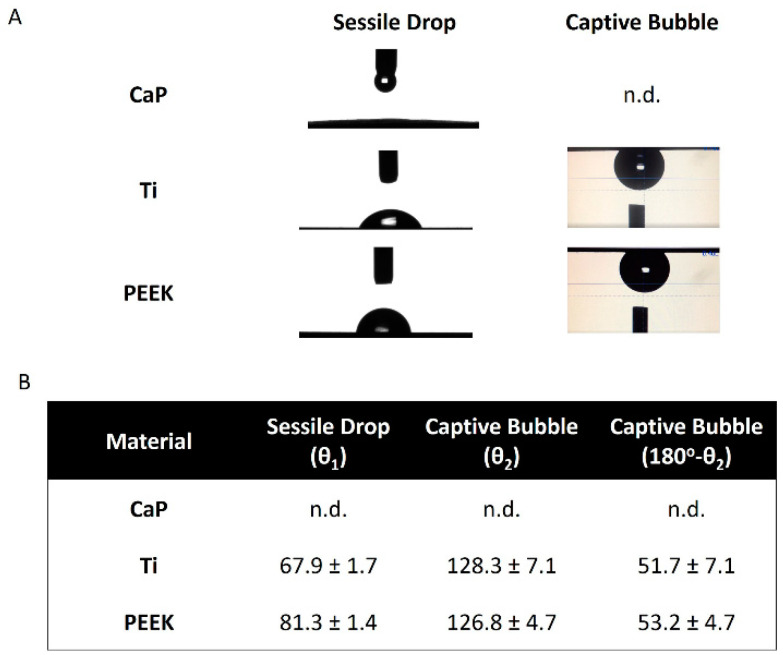
Assessment of wettability for the investigated materials. (**A**) Water contact angle measurements for calcium phosphate (CaP), titanium alloy (Ti), and polyetheretherketone (PEEK) obtained through sessile and captive methods. (**B**) Corresponding contact angle presented as the mean ± SD (*n* = 3). n.d.—not determined because of being experimentally inaccessible (see the text for further information).

**Figure 4 jfb-14-00361-f004:**
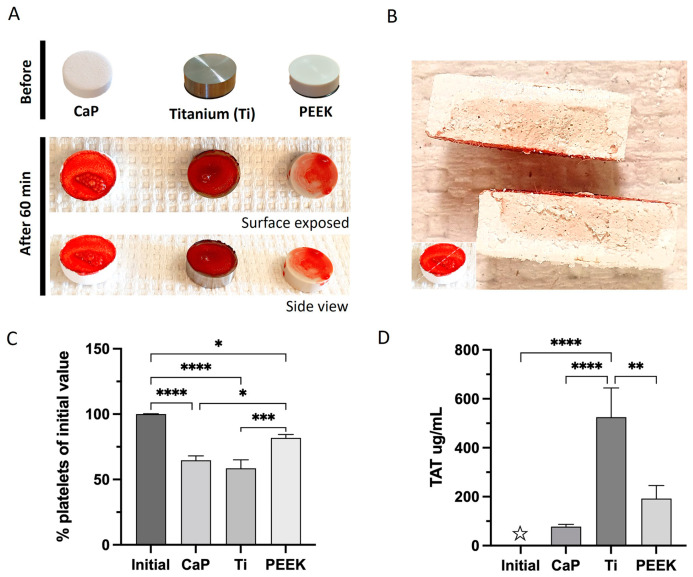
Coagulation activation after 60 min of contact with the calcium phosphate (CaP), titanium alloy (Ti), and polyetheretherketone (PEEK) material discs. (**A**) Representative macroscopic images of adherent blood cells and/or components with the surface of the materials. (**B**) Sectioning of CaP after 60 min of blood contact. (**C**) Percentage of platelets remaining in terms of initial values measured prior to contact at time 0 min. (**D**) The thrombin–antithrombin complex (TAT) compared to the initial amount (☆—the initial values of TAT are above zero, but due to the scale range applied, it is not visible). **** *p* < 0.0001, *** *p* < 0.001, ** *p* < 0.01, * *p* < 0.05; ANOVA with Tukey’s multiple comparisons test. Data are the average value of two wells/donor represented as the mean ± SEM (*n* = 13). All comparisons not presented are non-significant.

**Figure 5 jfb-14-00361-f005:**
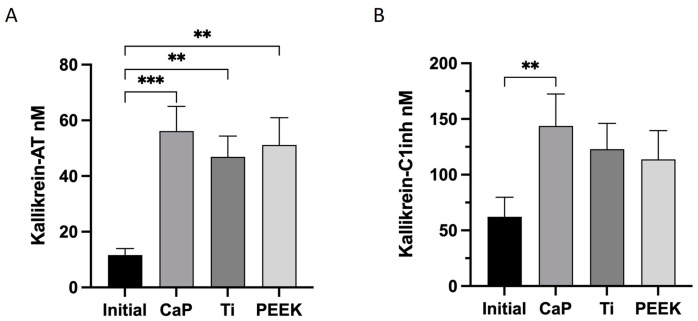
Activation of kallikrein–bradykinin (KK) system mediators after 60-min contact with the calcium phosphate (CaP), titanium alloy (Ti), and polyetheretherketone (PEEK) material discs. (**A**) The kallikrein (KK)–antithrombin (AT) complex. (**B**) The KK–C1–esterase inhibitor complex (C1- C1Inh). *** *p* < 0.001; ** *p* < 0.01. ANOVA with Tukey’s multiple comparisons test. The data are the average value of two wells/donor represented as the mean ± SEM (*n* = 12). All comparisons not presented are non-significant.

**Figure 6 jfb-14-00361-f006:**
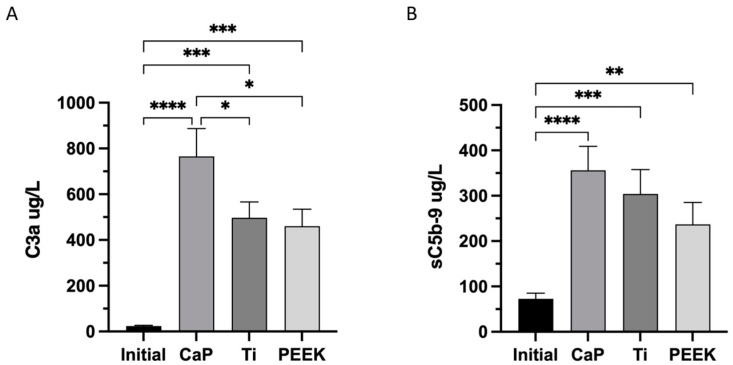
Complement system activation after 60 min of contact with the calcium phosphate (CaP), titanium alloy (Ti), and polyetheretherketone (PEEK) material discs. (**A**) Quantification of complement component 3 (C3a). (**B**) Plasma terminal C5b-9 complement complex. **** *p* < 0.0001, *** *p* < 0.001, ** *p* < 0.01, and * *p* < 0.05. ANOVA with Tukey’s multiple comparisons test. Data are the average value of two wells/donor represented as the mean ± SEM (*n* = 13). All comparisons not presented are non-significant.

**Figure 7 jfb-14-00361-f007:**
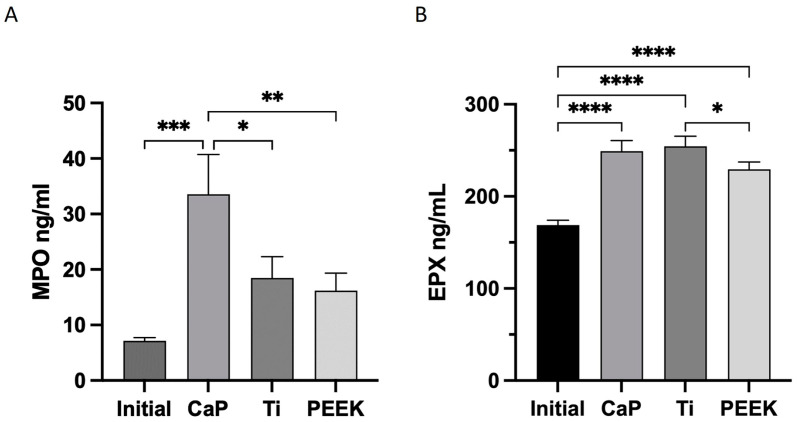
Human peroxidase activity after 60 min of contact with the calcium phosphate (CaP), titanium alloy (Ti), and polyetheretherketone (PEEK) material discs. (**A**) Quantification of myeloperoxidase (MPO) and (**B**) quantification of human eosinophil peroxidase (EPX). **** *p* < 0.0001, *** *p* < 0.001, ** *p* < 0.01, and * *p* < 0.05. ANOVA with Tukey’s multiple comparisons test. Data represent the average value of two wells/donor represented as the mean ± SEM (*n* = 10). All comparisons not presented are non-significant.

**Figure 8 jfb-14-00361-f008:**
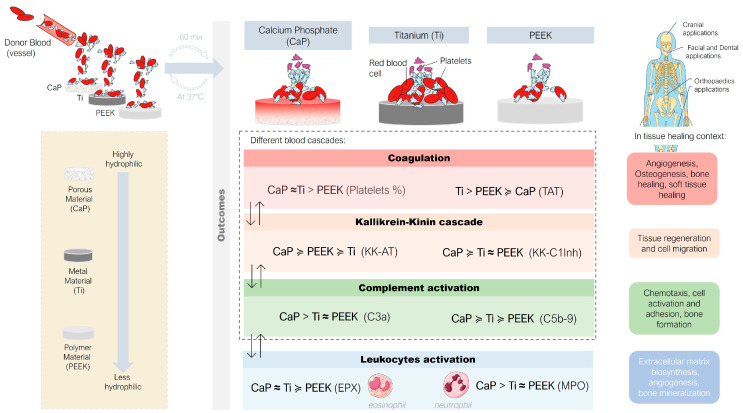
Schematic overview of local interactions between the different components of blood cascades and leukocyte responses during the early stages of contact with calcium phosphate (CaP), titanium alloy (Ti), and polyetheretherketone (PEEK) materials. Further details can be found through the results and discussion. (The symbol almost equal to (≈) by itself or associated with the symbol greater (≥) indicates that the values were significant between the data groups).

## Data Availability

Due to confidentiality agreements, data can be made available subject to a non-disclosure agreement and from the corresponding author upon request.

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
