# Peer review of "Human Whole Blood Interactions with Craniomaxillofacial Reconstruction Materials: Exploring In Vitro the Role of Blood Cascades and Leukocytes in Early Healing Events"

_jfb, 2023, doi:10.3390/jfb14070361_

Round 1

Reviewer 1 Report

In this paper, the authors investigated early interactions between three alloplastic materials (calcium phosphate (CaP), titanium alloy (Ti) and polyetheretherketone (PEEK) with human whole blood, using an established in vitro slide chamber model. Contrary to previous concepts emphasizing strong coagulation and platelet activation for healing post-implantation of metals, alloys, and plastics, the blood response of CaP material suggests that complement activation may be crucial for tissue healing. The reviewer thinks this paper may be an interesting research work, and minor revision is needed before accepted by JBF.

Some comments are given as follows:

1) In the Introduction section, the reference citing should be modified. Such as Line 46, the refs. should be behind the full stop. Line 51, Line 53, Line 55…

2) In this study, “we focus on the initial stages of material-blood interaction by exposing three miniaturized replicas of craniomaxillofacial implants constructed from polyether ether ketone (PEEK), titanium alloy (Ti), and a triphasic calcium phosphate (CaP) to freshly collected human whole blood in an established in vitro slide model.” The reviewer suggested that the previous studies on interactions between the PEEK, Ti and CaP and blood should be overviewed.

3) The A, B, and C should be added on the responding pictures. At the same time, Fig. 1 was also a part of Fig. 8. The reviewer suggested that the appropriate modifications are needed.

4) In Material Characterization section, Lines 89-97 as well as Lines 114-122 should be one paragraph.

5) Line 21, (PEEK)) should be (PEEK). The similar grammar errors should be revised.

Minor editing of English language required.

Author Response

Manuscript ID: jfb-2476976

Response to the Reviewers

Reply to reviewer #1:

Reviewer´s Comments to the Authors:

  • In the Introduction section, the reference citing should be modified. Such as Line 46, the refs. should be behind the full stop. Line 51, Line 53, Line 55…

Authors' response:   We thank you for the feedback regarding the formatting error. We have made amendments to the reference citing, ensuring that all references are now positioned following the journal outline.

  • In this study, “we focus on the initial stages of material-blood interaction by exposing three miniaturized replicas of craniomaxillofacial implants constructed from polyether ether ketone (PEEK), titanium alloy (Ti), and a triphasic calcium phosphate (CaP) to freshly collected human whole blood in an established in vitro slide model.” The reviewer suggested that the previous studies on interactions between the PEEK, Ti and CaP and blood should be overviewed. 

Authors' response:  Our study references previous investigations involving Ti-testing with blood. We have reviewed the literature and could not locate any applicable publications where either PEEK or calcium phosphate have been subjected to whole blood testing in a comparable way. The only literature we found about PEEK is regarding tests with platelet-rich plasma which is not comparable to whole blood used in our study.

  • The A, B, and C should be added on the responding pictures. At the same time, Fig. 1 was also a part of Fig. 8. The reviewer suggested that the appropriate modifications are needed.

Authors response: We thank you for the reviewer's comment. We have updated Figure 1 to include parts A, B, and C. We agree with the reviewer and removed Figure 1 from Figure 8 and rearranged figure 8, accordingly.

  • In Material Characterization section, Lines 89-97 as well as Lines 114-122 should be one paragraph.

Authors’ response:  Adjustments have been made to the sections, and mentioned Lines are now only one paragraph as suggested.

  • Line 21, (PEEK)) should be (PEEK). The similar grammar errors should be revised.

Authors’ response: We thank you for the reviewer's comment. The incorrect parentheses have been deleted and the manuscript has been revised for similar grammar/editing errors.

Reviewer 2 Report

  1. The title of the submitted manuscript could be misleading, given that it failed to explain the study materials and results adequately. It involves the term “cranio-maxillofacial reconstruction,” which is not conducted in this study. Also, the in vitro nature of this study should be clearly stated in the title to avoid potential assumptions about the clinical nature of the study. 
  2. The authors stated that “three miniaturized replicas of craniomaxillofacial implants” were used in the manuscript (line 70), which could be misleading considering that in methodology, materials were described as “disc-shaped with a diameter of 16 mm and a height of 5 mm.” (line 80)
  3. In the results, some findings on Scanning electron microscopy are described as a consequence of cutting and polishing (lines 228, 229), while the cutting and polishing procedure was not described in the methodology section (lines 89-98).
  4. It would be beneficial to state the magnification used for SEM imaging (in the text, Figure 1, or both).
  5. The authors do not specify the number of blood donors and which blood was used for which material (lines 116 and 117). This is an important issue to address and properly declare if any difference could influence the results. 
  6. It is unexpected that the first mentioned figure is labeled as Figure 3 (line 126)
  7. The author stated that CaP discs were the only ones that were sectioned to examine the internal structure, while it would be beneficial to offer an explanation why this was not done in the other two materials (line 160).
  8. Line 220: It seems that “surface morphology or surface topography” would be a more suitable term than “surface structure.”
  9. Line 255: I wonder if there is any data about contact angle values in titanium alloy?
  10. The authors should check whether all of the data included in the study are properly stated using proper standard units. The standard deviation should have the same unit as the stated means.
  11. From the graphic representation in Figure 5, the increase was not precisely three times the initial value, as stated in line 301. Rephrasing the sentence to properly and accurately describe the data is highly advisable.
  12. Line 326: It needs to be clarified and should be explained why this manuscript focused only on two types of leukocytes (neutrophils and eosinophils).
  13. It would be beneficial for the readers if, in the discussion or in the conclusion section, the potential clinical relevance of this data is stated or to point out the directions to broader the research area to have clinical relevance.

Minor editing of the English language is required.

Author Response

Response to the Reviewers

Reply to reviewer #2:

Reviewer´s Comments to the Authors:

  1. The title of the submitted manuscript could be misleading, given that it failed to explain the study materials and results adequately. It involves the term “cranio-maxillofacial reconstruction,” which is not conducted in this study. Also, the in vitro nature of this study should be clearly stated in the title to avoid potential assumptions about the clinical nature of the study. 

Authors' response: We appreciate your feedback regarding the potential uncertainty of the original title. In response, we have revised it for greater clarity to accurately represent our study. The new title is: "Human Whole Blood Interactions with Craniomaxillofacial Reconstruction materials: Exploring in vitro the Role of Blood Cascades and Leukocytes in Early Healing Events".

  1. The authors stated that “three miniaturized replicas of craniomaxillofacial implants” were used in the manuscript (Line 70), which could be misleading considering that in methodology, materials were described as “disc-shaped with a diameter of 16 mm and a height of 5 mm.” (Line 80)

Authors’ response: We agree that the descriptor "miniaturized" does not accurately represent the disc used in the study, and as such, we have decided to omit the term and rephrase the sentence, which now reads: “In this study, we focus on the initial stages of material-blood interaction by exposing three disc-shaped replicas of craniomaxillofacial implants constructed from polyether ether ketone (PEEK), titanium alloy (Ti), and a triphasic calcium phosphate (CaP) to freshly collected human whole blood in an established in vitro slide model.” (Line 70).

  1. In the results, some findings on Scanning electron microscopy are described as a consequence of cutting and polishing (Lines 228, 229), while the cutting and polishing procedure was not described in the methodology section (Lines 89-98).

Author’s response: We thank you for the comment. The methodology section was updated to become clearer, and the following sentence was included: “The discs made from Ti and PEEK discs were cut from rods of the same diameter, polished with silicon carbide paper, and then subjected to an ultrasonic cleaning procedure to remove any residual particles.” (Line 82-84).

  1. It would be beneficial to state the magnification used for SEM imaging (in the text, Figure 1, or both).

Authors' response: We thank you for this valuable comment. The magnification used in the SEM imaging has been included in the Figure caption (Line 236-238).

  1. The authors do not specify the number of blood donors and which blood was used for which material (Lines 116 and 117). This is an important issue to address and properly declare if any difference could influence the results. 

Authors’ response: The number of blood donors has been incorporated into the text, which now reads: "In brief, we collected 1.3 ml of blood from seven healthy donors, and this was transferred to a two-well heparinized slide chamber identical in diameter to the prepared materials discs. Every material tested was subjected to the blood from each individual donor." (Line 116-118).

  1. It is unexpected that the first mentioned figure is labeled as Figure 3 (Line 126)

Authors’ response: Thank you for pointing this out. We have now repositioned Figure 3 to be Figure 1, and the previous Figures 1 and 2 have been renamed to Figures 2 and 3, respectively.

  1. The author stated that CaP discs were the only ones that were sectioned to examine the internal structure, while it would be beneficial to offer an explanation why this was not done in the other two materials (Line 160).

Authors’ response: We understand the reviewer's comment however, given that PEEK and Ti are solid materials, reactions are expected to occur on their surface. In addition, sectioning of CaP was performed to verify that the change of external color observed in the disc was also spreading into the internal structure.  

  1. Line 220: It seems that “surface morphology or surface topography” would be a more suitable term than “surface structure.”

Authors’ response: We have replaced the term "surface structure" with "surface topography" (Line 232-233).

  1. Line 255: I wonder if there is any data about contact angle values in titanium alloy?

Authors’ response: We thank you for the valuable comment. There are published studies with contact angle values for titanium that highlight that the contact angle for a polished titanium cranioplasty plate can depend on many factors including the specific polishing process and the cleanliness of the surface.  We have rephrased the sentence and it now reads: “In literature, the contact angle of Ti alloy with different topographic finishes and untreated PEEK is between 30 -70° and 70°- 90°, respectively [25,26].” (Lines 273-275).

  1. The authors should check whether all of the data included in the study are properly stated using proper standard units. The standard deviation should have the same unit as the stated means.

Authors’ response: The data are presented using standard units or those suggested by the ELISA-kit manufacturers, except for platelets, as detailed in the manuscript. However, we recognize that we were inconsistent in placing the standard unit throughout the document. We have revised the manuscript (Lines 336-337) and now the standard units should be coherent throughout the manuscript.

  1. From the graphic representation in Figure 5, the increase was not precisely three times the initial value, as stated in Line 301. Rephrasing the sentence to properly and accurately describe the data is highly advisable.

Authors’ response: We agree that the initial statement lacked accuracy and have thus updated it to state: “the CaP surface showed the most pronounced increase in both complexes.” (Line 307).

  1. Line 326: It needs to be clarified and should be explained why this manuscript focused only on two types of leukocytes (neutrophils and eosinophils).

Authors’ response: We thank you for the comment and understand the reviewer’s comment. The analysis of release products from neutrophils and eosinophils was added as a result of complement activation measurements after the end of the experimental setup and literature search. No additional measurements were possible. We address the comment by adding further limitations in the discussion (Line 521-522) and clarifying it in the results sub-section (Line 332-333).

  1. It would be beneficial for the readers if, in the discussion or in the conclusion section, the potential clinical relevance of this data is stated or to point out the directions to broader the research area to have clinical relevance.

Authors’ response:  We thank you for this valuable comment. Following the reviewer's suggestion, we have revised the last sentence of the discussion and conclusion, and now reads:

“Future studies should aim to explore how the inflammatory response to implantable materials is modulated over time, concentrating on the recruitment of various specific cells and their differentiation at various time points.” (Line 520-523).

“These findings offer insights into blood-material interactions and could guide the further development and optimization of existing implantable materials used in today's clinical practice. By modulating the initial immune response, the healing process could be optimized, possibly resulting in fewer complications and better osteointegration of implant materials.” (Line 529-533).

Round 2

Reviewer 2 Report

I thank the authors for addressing all reviewers' concerns. The substantial changes were incorporated to reflect most of the reviewer's suggestions and subsequently to improve the quality of this manuscript.  I have no further major comments or concerns about this manuscript.

As probably in any manuscript written by non-native English speakers, some minor language editing would be beneficial to help readers understand the content better. However, the manuscript is understandable in its present form, so the additional language editing is not crucial.